# Evaluation of chlorogenic acid and carnosol for anti-efflux pump and anti-biofilm activities against extensively drug-resistant strains of *Staphylococcus aureus* and *Pseudomonas aeruginosa*

Mohaddeseh Sheikhy,[1] Vajihe Karbasizade,[1] Mustafa Ghanadian,[2] Hossein Fazeli[1]

**ABSTRACT**  Efflux pumps and biofilm play significant roles in bacterial antibiotic resistance. This study investigates the potential of chlorogenic acid (CGA) and carnosol (CL), as phenolic and diterpene compounds, respectively, for their inhibitory effects on efflux pumps. Among the 12 multidrug-resistant (MDR) strains of *Staphylococcus aureus* and *Pseudomonas aeruginosa* isolated from nosocomial skin infections, eight strains were identified as extensively drug resistant (XDR) using the disc diffusion method. The presence of efflux pumps in MDR strains of *S. aureus* and *P. aeruginosa* was screened using carbonyl cyanide-m-chlorophenylhydrazone. Between the 12 MDR strains of *S. aureus* and *P. aeruginosa*, 80% (4 out of 5) of the *S. aureus* strains and 85.7% (6 out of 7) of the *P. aeruginosa* strains exhibited active efflux pumps associated with gentamicin resistance. The checkerboard assay results, in combination with gentamicin, demonstrated that CGA exhibited a reduction in the minimum inhibitory concentration (MIC) for XDR *S. aureus* strain. Similarly, CL showed a synergistic effect and reduced the MIC for both XDR strains of *S. aureus* and *P. aeruginosa*. Flow cytometry was used to examine efflux pump activity at sub-MIC concentrations of 1/8, 1/4, and 1/2 MIC in comparison to the control. In XDR *S. aureus*, CGA demonstrated 39%, 70%, and 19% inhibition, while CL exhibited 74%, 73.5%, and 62% suppression. In XDR *P. aeruginosa*, CL exhibited inhibition rates of 25%, 10%, and 15%. The inhibition of biofilm formation was assessed using the microtiter plate method, resulting in successful inhibition of biofilm formation. Finally, the MTT assay was conducted, and it confirmed minimal cytotoxicity. Given the significant reduction in efflux pump activity and biofilm formation observed with CGA and CL in this study, these compounds can be considered as potential inhibitors of efflux pumps and biofilm formation, offering potential strategies to overcome antimicrobial resistance.

**IMPORTANCE**  In summary, CGA and CL demonstrated promising potentiating antimicrobial effects against XDR strains of *Staphylococcus aureus* and *Pseudomonas aeruginosa*, suggesting their probably potential as candidates for addressing nosocomial pathogens. They exhibited significant suppression of efflux pump activity, indicating a possible successful inhibition of this mechanism. Moreover, all substances effectively inhibited biofilm formation, while showing minimal cytotoxicity. However, further advancement to clinical trials is needed to evaluate the feasibility of utilizing CGA and CL for reversing bacterial XDR efflux and determining their efficacy against biofilms. These trials will provide valuable insights into the practical applications of these compounds in combating drug-resistant infections.

**KEYWORDS**  *Staphylococcus aureus*, *Pseudomonas aeruginosa*, chlorogenic acid, carnosol, antimicrobial resistance, nosocomial infection

Address correspondence to Hossein Fazeli, h_fazeli@med.mui.ac.ir.

The authors declare no conflict of interest.

See the funding table on p. 15.

Antimicrobial resistance (AMR) indeed poses a critical and urgent global challenge. The overuse, inappropriate prescription, and extensive use of antibiotics in veterinary and agricultural settings are major contributors to the emergence and spread of AMR (1). According to the 2019 Global Burden of Disease Study, several bacterial pathogens, including *Staphylococcus aureus*, *Escherichia coli*, *Streptococcus pneumoniae*, *Klebsiella pneumoniae*, and *Pseudomonas aeruginosa*, are responsible for a significant proportion of bacterial-related deaths worldwide (2).

*S. aureus* is a ubiquitous and opportunistic pathogen that has the potential to induce a wide range of illnesses, from skin infections to severe conditions such as endocarditis, osteomyelitis, and sepsis, which can have mild to life-threatening implications (3).

*P. aeruginosa* is a gram-negative opportunistic bacterium commonly associated with nosocomial infections. Multidrug-resistant (MDR) *P. aeruginosa* is a major cause of death in burn patients, accounting for 64% of cases, with MDR organisms being responsible for 86% of sepsis-related deaths in pediatric burn intensive care units (BICUs) (4).

The loss of the skin's natural defenses against infection and the resulting immunosuppression make burn patients more susceptible to infection. Both in the burn intensive care unit and burn common wards (BCW), gram-negative bacteria were the main pathogens in patients with burns. The most common pathogens in BICU were *Acinetobacter baumannii*, *S. aureus*, and *P. aeruginosa*, whereas the most prevalent pathogens in BCW were *S. aureus*, *P. aeruginosa*, and *A. baumannii*. In comparison to BCW, the overall MDR rate was higher in BICU. Gentamicin, a commonly used antibiotic for burn infection treatment, exhibited varying resistance rates. Gentamicin resistance in BICU and BWC for *P. aeruginosa* and *S. aureus* ranged from 42.5% to 13.5% and 94.4% to 65.3%, respectively (5).

One of the fundamental mechanisms of antibiotic resistance in bacteria is the overexpression of multidrug efflux pumps, which were considered important in gentamicin resistance. Efflux pumps are a group of proteins found in the bacterial cell membrane that actively export toxic substances, including antimicrobial agents, and quorum-sensing (QS) signal molecules that promote antibiotic resistance and biofilm formation. They have the capability to export various substrates with different structural characteristics or specialize in a single substrate. The association between extensively drug-resistant (XDR) bacteria and efflux pumps, which have the ability to export multiple substrates, including various classes of antibiotics, has been observed (6, 7). Considering that the active efflux of antibacterial agents plays a significant role in bacterial antibiotic resistance and serves as the primary line of defense against antimicrobial agents, the inhibition of efflux pumps holds promise as a method to restore the potency of antibacterial agents (8, 9).

The potential use of efflux pump inhibitors (EPIs) as therapeutic agents to restore the antibacterial activity of antibiotics that have lost some efficacy has been suggested (10). However, currently, no clinically approved medication exists specifically for EPIs (11).

In recent years, there has been a growing interest in exploring natural products for the discovery of antimicrobial compounds (12). Natural products offer several advantages, including high stereochemistry, diverse scaffolds (rings), and low toxicity, making them a potentially valuable source of compounds for antimicrobial discovery and development (13).

Phenolic or polyphenolic compounds are one of the most prevalent groups of secondary metabolites. Chlorogenic acid (CGA), also known as coffee tannic acid or 3-caffeoylquinic acid, is a phenyl acrylate polyphenol compound synthesized from quinic and caffeic acid (14). Another important group of natural secondary metabolites with significant antimicrobial activity is diterpenoids (15). Among them, carnosol (CL; picrosalvin) stands out as an oxidative diterpene derived from carnosic acid. Carnosol has exhibited anti-inflammatory, antibacterial, anticarcinogenic, and antioxidant activity in both *in vitro* and *in vivo* experimental models (16).

The work conducted in this study holds significant importance, particularly in the evaluation of antimicrobial modulation, efflux pump inhibition, and antibiofilm assays

utilizing phenolic and diterpenic compounds. Accordingly, the objective of this study was to investigate the antimicrobial activities of chlorogenic acid and carnosol in combination with gentamicin using *in vitro* assays against extensively drug-resistant strains of *S. aureus* and *P. aeruginosa*.

## RESULTS

### Antibacterial susceptibility testing results

As mentioned earlier, antibiotic susceptibility of all clinical strains of *S. aureus* and *P. aeruginosa* isolated from human skin, responsible for wound and burn infections, was assessed using the disk diffusion method. Susceptible (S), intermediate (I), and resistant (R) strains were identified based on MIC breakpoints according to references (17, 17), as demonstrated in Tables 1 and 2. Strains that are susceptible to only one or two antibiotics from various classes are categorized as XDR. Gentamicin MIC value was also determined for all clinical strain using microplate broth microdilution method. Clinical strains had different gentamicin MIC values that ranged from 8 to 512 for *S. aureus* and 2 to 1,024 µg/mL for *P. aeruginosa*, respectively. The gentamicin MIC distributions for all strains of *S. aureus* and *P. aeruginosa* are listed in Tables 1 and 2. The most resistant XDR strain of each bacterium was used for subsequent steps (*S. aureus*-3, and *P. aeruginosa*-5).

### Antibacterial potentiating effect

The antibacterial potentiating effect of chlorogenic acid and carnosol was evaluated. None of the substances exhibited clinically applicable antibacterial activity, as MIC values were above or equal to 1,024 µg/mL. However, when combined with gentamicin in checkerboard assays, CGA demonstrated a 2-log decrease in the MIC of gentamicin against selected XDR *S. aureus*-3 (512–128 µg/mL, FICI = 0.3), indicating a synergistic effect. On the other hand, CGA had an additive effect on selected XDR *P. aeruginosa*-5 (1,024–512 µg/mL, FICI = 0.75). CL resulted in a 4-log reduction in gentamicin MIC against XDR *S. aureus*-3 (512–32 µg/mL, FICI = 0.125) and a 1-log reduction against XDR *P. aeruginosa*-5 (1,024–512 µg/mL, FICI = ~0.5), demonstrating a synergistic effect on both studied XDR strains. Detailed results are presented in Table 3.

### Efflux pump screening

Carbonyl cyanide-m-chlorophenylhydrazone (CCCP) did not demonstrate any significant antibacterial activity on its own, as the MIC values for studied strains were higher than 1,024 µg/mL. For the efflux pump screening, the application of CCCP in combination with gentamicin resulted in a reduction in the MIC of gentamicin in many of the tested clinical XDR strains (Tables 1 and 2). The findings indicated that among the gentamicin-resistant XDR strains of *S. aureus* and *P. aeruginosa*, 60% (3 out of 5) and 50% (2 out of 4) of the

**TABLE 1** Kirby-Bauer disk diffusion susceptibility results for all clinical strains of *S. aureus*[a]

| Bacteria | Interpretation of Kirby-Bauer disk diffusion susceptibility test | | | | | | | | | MICs of gentamicin | MICs of gentamicin after CCCP treatment |
|---|---|---|---|---|---|---|---|---|---|---|---|
| | CIP | GEN | CLH | DOX | FOX | PEN | SXT | ERY | CLN | | |
| *S. aureus* (ATCC 25923) | S | S | S | S | S | S | S | S | S | –[b] | – |
| *S. aureus*-1 | R | R | S | R | R | R | S | R | R | 32 µg/mL | 2 µg/mL |
| *S. aureus*-2 | R | R | S | R | R | R | S | R | R | 32 µg/mL | 4 µg/mL |
| *S. aureus*-3 | R | R | S | R | R | R | S | R | R | 512 µg/mL | 32 µg/mL |
| *S. aureus*-4 | R | I | S | S | R | R | S | R | R | 8 µg/mL | 8 µg/mL |
| *S. aureus*-5 | R | R | S | S | R | R | S | R | R | 16 µg/mL | 1 µg/mL |

[a]CCCP, carbonyl cyanide-m-chlorophenylhydrazone; CIP, ciprofloxacin; GEN, gentamicin; CLH, chloramphenicol; DOX, doxycycline; FOX, cefoxitin; PEN, penicillin; SXT, trimethoprim/sulfamethoxazole; ERY, erythromycin; CLN, clindamycin; S, susceptible; I, intermediate; R, resistant.
[b] –, the standard strain of *Staphylococcus aureus* (ATCC 25923) was sensitive to the antibiotics under investigation.

**TABLE 2** Kirby-Bauer disk diffusion susceptibility results for all clinical strains of *P. aeruginosa*[a]

| Bacteria | Interpretation of Kirby-Bauer disk diffusion susceptibility test | | | | | | MICs of gentamicin | MICs of gentamicin after CCCP treatment |
|---|---|---|---|---|---|---|---|---|
| | CIP | GEN | FEP | PTZ | CAZ | MER | | |
| *P. aeruginosa* (ATCC 27853) | S | S | S | S | S | S | – | – |
| *P. aeruginosa*-1 | R | R | R | R | R | R | 256 µg/mL | 16 µg/mL |
| *P. aeruginosa*-2 | R | S | R | R | R | R | 4 µg/mL | 4 µg/mL |
| *P. aeruginosa*-3 | R | R | I | R | S | R | 512 µg/mL | 256 µg/mL |
| *P. aeruginosa*-4 | R | R | R | R | R | R | 512 µg/mL | 128 µg/mL |
| *P. aeruginosa*-5 | R | R | R | R | R | R | 1,024 µg/mL | 32 µg/ml |
| *P. aeruginosa*-6 | R | S | I | S | S | S | 2 µg/mL | 1 µg/mL |
| *P. aeruginosa*-7 | R | S | R | R | S | S | 4 µg/mL | 1 µg/mL |

[a]CCCP, carbonyl cyanide-m-chlorophenylhydrazone; CIP, ciprofloxacin; GEN, gentamicin; CLH, chloramphenicol; DOX, doxycycline; FOX, cefoxitin; PEN, penicillin; SXT, trimethoprim/sulfamethoxazole; ERY, erythromycin; CLN, clindamycin; S, susceptible; I, intermediate; R, resistant.

strains, respectively, exhibited a fourfold decrease in their gentamicin MIC. This decrease in MIC indicated the presence of active gentamicin resistance efflux pumps in these strains, classifying them as strong efflux pump strains.

## Efflux pump inhibition assay

To investigate efflux pump inhibition, the most resistant strain of each bacterium, which displayed the most significant decrease in its gentamicin MIC following CCCP utilization, was selected for further analysis using flow cytometry.

### Flow cytometric analysis of efflux pump inhibition on the XDR S. aureus-3

All experiments conducted in this study utilized concentrations of gentamicin, CGA, and CL that did not impact cell viability. To investigate the efflux pump activity, subinhibitory concentrations (1/8, 1/4, and 1/2 MIC) of gentamicin in combination with sub-MIC concentrations of CGA (16, 32, and 64 µg/mL along with 8, 16, and 32 µg/mL) and CL (4, 8, and 16 µg/mL along with 8, 16, and 32 µg/mL) were evaluated, respectively. Control experiments without CGA and CL were also performed using the same concentrations of gentamicin to assess their effect on efflux pump activity. Two-dimensional plots associated with flow cytometry experiments were constructed, as illustrated in Fig. 1 and 2. The analysis boundary was determined using the forward scatter and side scatter plot based on unstained cells. The four quadrants (Q1–Q4) were defined by placing the majority of dots in the Q4 region of the unstained sample. The emission range of Rh 123 (FL-2 channel) was used to determine the specific percentage of cells associated with efflux pump activity. Comparing the treatment groups with the corresponding controls, CGA exhibited 39%, 70%, and 19% improvement in efflux pump inhibition at the mentioned concentrations (Fig. 4a). In the case of CL, an approximately 74%, 73.5%, and 62% progression in efflux pump inhibition was observed when comparing the treatment group with the controls (Fig. 4b). The most significant effects are summarized in Table 4.

**TABLE 3** Assessment of gentamicin antimicrobial activity in combination with CGA and CL by checkerboard assay

| Bacteria | MICs of: | | | | | | | Checkerboard results based on interpretation of FIC index | |
|---|---|---|---|---|---|---|---|---|---|
| | GEN and compounds before treatment | | | GEN and compounds after treatment | | | | | |
| | GEN | CGA | CL | GEN + CGA | GEN + CL | CGA | CL | GEN + CGA | GEN + CL |
| XDR *S. aureus*-3 | 512 µg/mL | 1,024 µg/mL | 1,024 µg/mL | 128 µg/mL | 32 µg/mL | 64 µg/mL | 64 µg/mL | Synergism (0.312) | Synergism (0.125) |
| XDR *P. aeruginosa*-5 | 1,024 µg/mL | 1,024 µg/mL | 1,024 µg/mL | 512 µg/mL | 512 µg/mL | 256 µg/mL | 16 µg/mL | Additive (0.750) | Synergism (~0.5) |

[a]GEN, gentamicin; FIC, fractional inhibitory concentration.

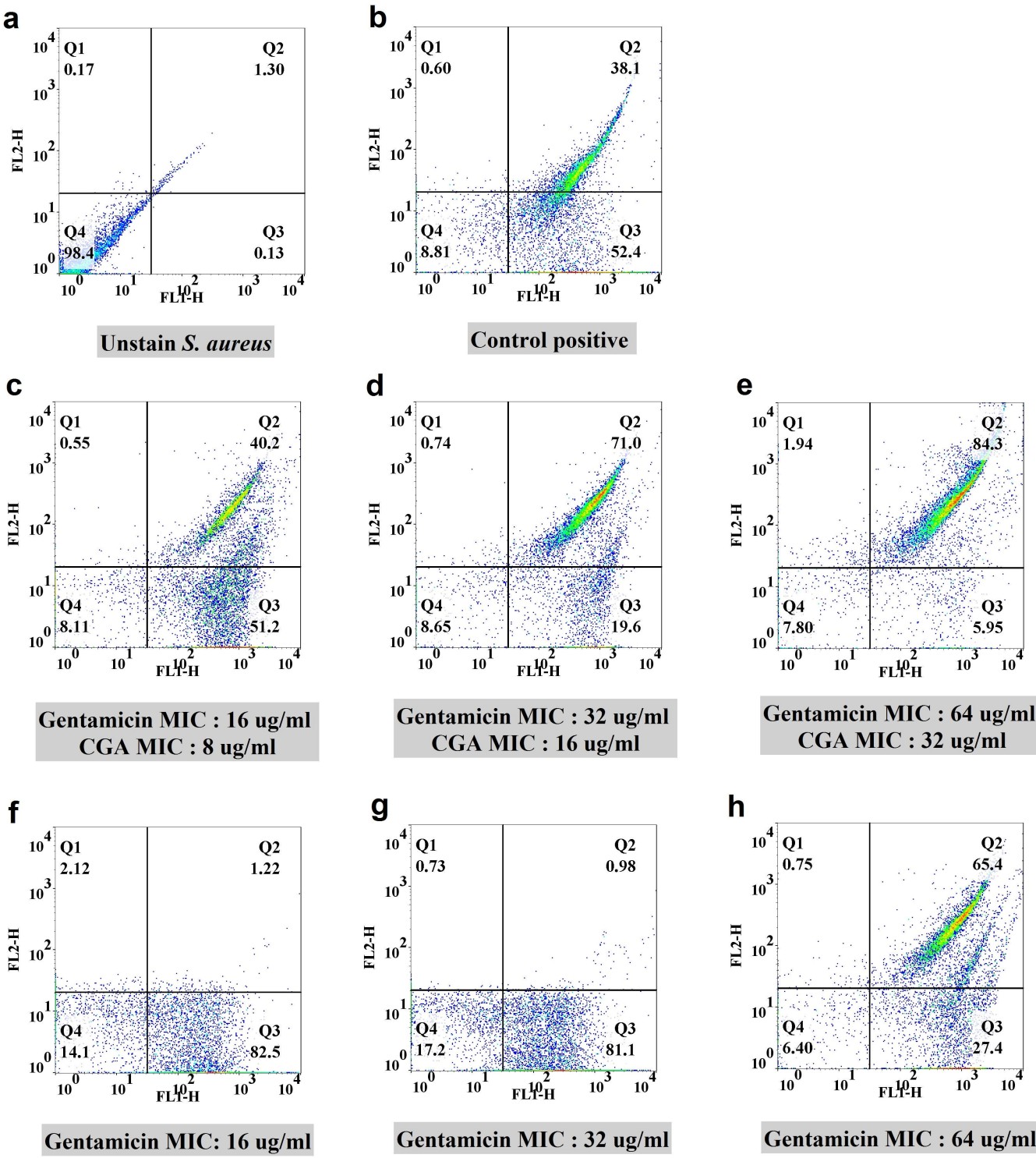

**FIG 1** Flow cytometric analysis of efflux pump inhibition by CGA on the selected XDR *S. aureus*. Two-dimensional plots for (a) unstained sample and (b) positive control containing CCCP. (c, d, and e) Sub-inhibitory concentrations (1/8, 1/4, 1/2 MIC) of gentamicin-CGA and (f, g, and h) gentamicin at the same concentrations as controls.

## Flow cytometric analysis of efflux pump inhibition on the XDR *P. aeruginosa*-5

Since CGA did not exhibit a synergistic effect on XDR *P. aeruginosa*, further experiments were conducted using subinhibitory concentrations (1/8, 1/4, and 1/2 MIC) of gentamicin

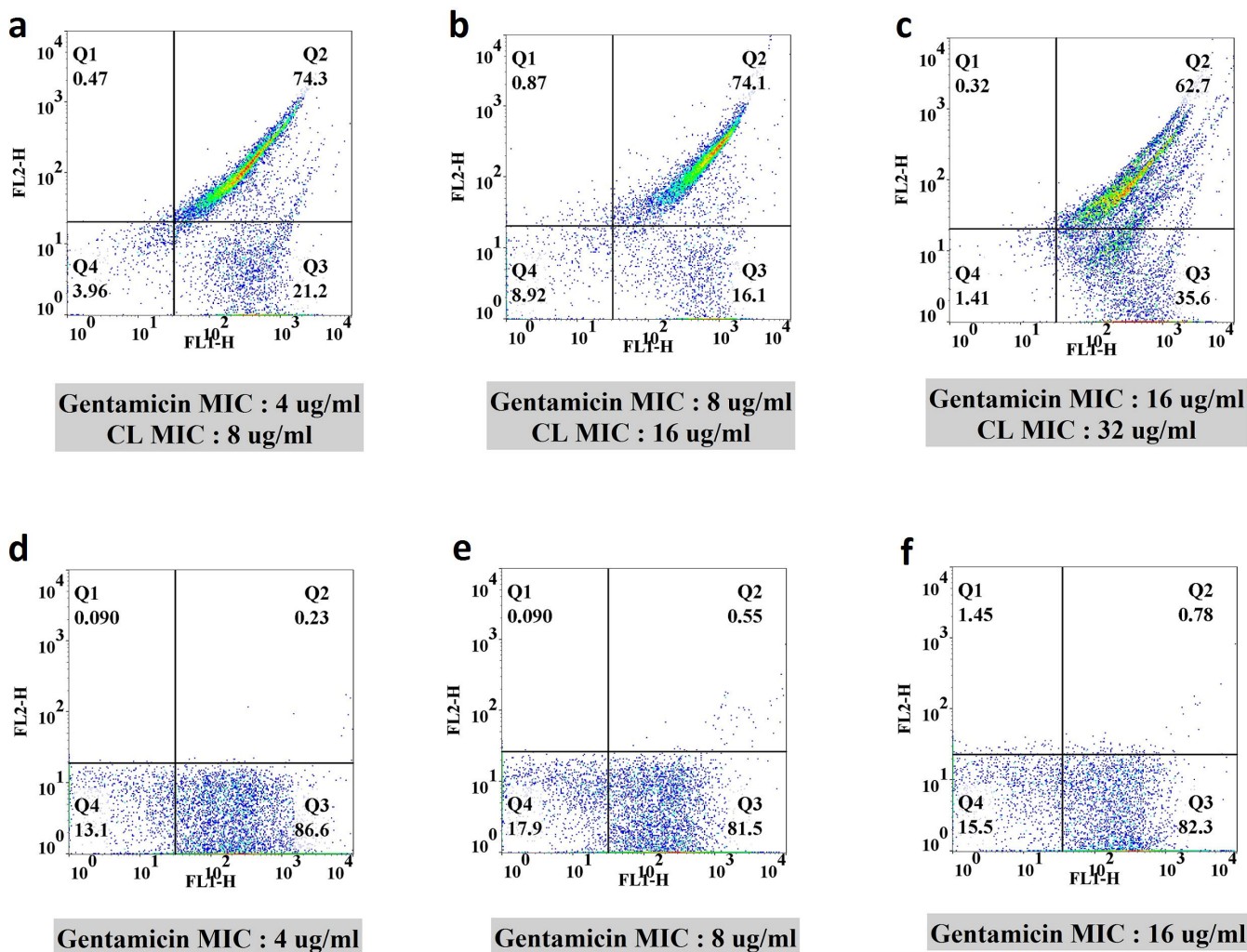

**FIG 2** Flow cytometric analysis of efflux pump inhibition by CL on the selected XDR *S. aureus*. (a, b, and c) Sub-inhibitory concentrations (1/8, 1/4, 1/2 MIC) of gentamicin-CL and (d, e, and f) gentamicin at the same concentrations as controls.

(64, 128, and 256 µg/mL) in combination with sub-MIC concentrations of CL (2, 4, and 8 µg/mL). Controls consisting of gentamicin at the same concentrations without CL were included. Two-dimensional plots associated with flow cytometry experiments were generated (Fig. 3). The specific percentage of cells related to efflux pump activity was determined by comparing the treatment group with its corresponding control. As depicted in Fig. 3, there was approximately 25%, 10%, and 15% improvement in efflux pump inhibition when comparing the treatment group with the control (Fig. 4c). The results of this assay are summarized in Table 4.

## Inhibition of biofilm formation

The antibiofilm assay was conducted in triplicate for each compound at their respective sub-MIC concentrations. A negative control was used as a blank, and the results were reported as mean ± standard deviation (SD), as depicted in Fig. 5. In the case of XDR *S. aureus*-3, all studied compounds demonstrated an inhibitory effect on biofilm formation, although the effect was not particularly strong. At subinhibitory concentrations of 1/8, 1/4, and 1/2 MIC, the combination of gentamicin with CGA resulted in biofilm inhibition rates of 3%, 21%, and 23%, respectively. Similarly, the combination of gentamicin with CL exhibited biofilm inhibition rates of 22%, 11%, and 17% (Fig. 5a). However, for XDR *P. aeruginosa*-5, the results were more noteworthy. Almost all substances at sub-MIC

**TABLE 4** The results of flow cytometric analysis of efflux pump inhibition

| Bacteria | Sub-MIC concentrations (1/8 to 1/2 MIC) | In combination with GEN | Only GEN (control) |
|---|---|---|---|
| XDR *S. aureus* | | CGA treatment | |
| | GEN MIC = 16 µg/mL | 40.2% | 1.22% |
| | CGA MIC = 8 µg/mL | | |
| | GEN MIC = 32 µg/mL | 71% | 0.98% |
| | CGA MIC = 16 µg/mL | | |
| | GEN MIC = 64 µg/mL | 84.3% | 65.4% |
| | CGA MIC = 32 µg/mL | | |
| XDR *S. aureus* | | CL treatment | |
| | GEN MIC = 4 µg/mL | 74.3% | 0.23% |
| | CL MIC = 8 µg/mL | | |
| | GEN MIC = 8 µg/mL | 74.1% | 0.55% |
| | CL MIC = 16 µg/mL | | |
| | GEN MIC = 16 µg/mL | 62.7% | 0.78% |
| | CL MIC = 32 µg/mL | | |
| XDR *P. aeruginosa* | | CL treatment | |
| | GEN MIC = 64 µg/mL | 85% | 60.6% |
| | CL MIC = 2 µg/mL | | |
| | GEN MIC = 128 µg/mL | 75.8% | 66.1% |
| | CL MIC = 4 µg/mL | | |
| | GEN MIC = 256 µg/mL | 79.4% | 64.1% |
| | CL MIC = 8 µg/mL | | |

[a]GEN, gentamicin; MIC, minimum inhibitory concentration.

concentrations displayed an antibiofilm potential of over 50%. The combination of gentamicin with CGA inhibited biofilm formation by 62%, 75%, and 80%, while the combination of gentamicin with CL achieved biofilm inhibition rates of 59%, 63%, and 46% (Fig. 5b).

## Analysis of cell viability by the MTT assay

The results of cell viability were expressed as a percentage relative to the control. As mentioned earlier, the maximum MIC values for CGA and CL in combination with gentamicin were 64 µg/mL for XDR strains of *S. aureus*-3 and *P. aeruginosa*-5. Therefore, cell viability was measured at concentrations ranging from 2× MIC (128 µg/mL), MIC (64 µg/mL), to 1/2 MIC (32 µg/mL). For CGA, compared to the control, the results obtained with the microplate reader at 570 nm indicated that 63% of the cells remained viable at 2× MIC, 76% at MIC, and 79% at 1/2 MIC. Similarly, for CL, the cell viability values were 64%, 77%, and 88%, respectively. The cell viability results are illustrated in Fig. 6.

## DISCUSSION

To the best of our knowledge, this study is the first to specifically investigate the efflux pump inhibition and antibiofilm activities of CGA and CL against XDR strains of *S. aureus* and *P. aeruginosa*. Given the significant presence of clinical *S. aureus* and *P. aeruginosa* strains with overexpressed efflux pumps, our research has focused on exploring the potential of CGA and CL as efflux pump inhibitors. Efflux pumps have the capability to expel various toxic substances, including antimicrobial agents, metabolites, and quorum-sensing signal molecules. By inhibiting efflux pumps, EPIs can prevent the extrusion of these substances, leading to increased intracellular concentrations of antibiotics without the need for higher dosages in patients. Consequently, previously resistant bacteria may regain sensitivity to specific antibiotic substrates (18).

In a study conducted by Su et al., it was discovered that CGA affects the inner membrane of *P. aeruginosa* P1, leading to a reduction in lipopolysaccharide (LPS) levels, which are major components of the outer membrane (OM). Additionally, CGA inhibits

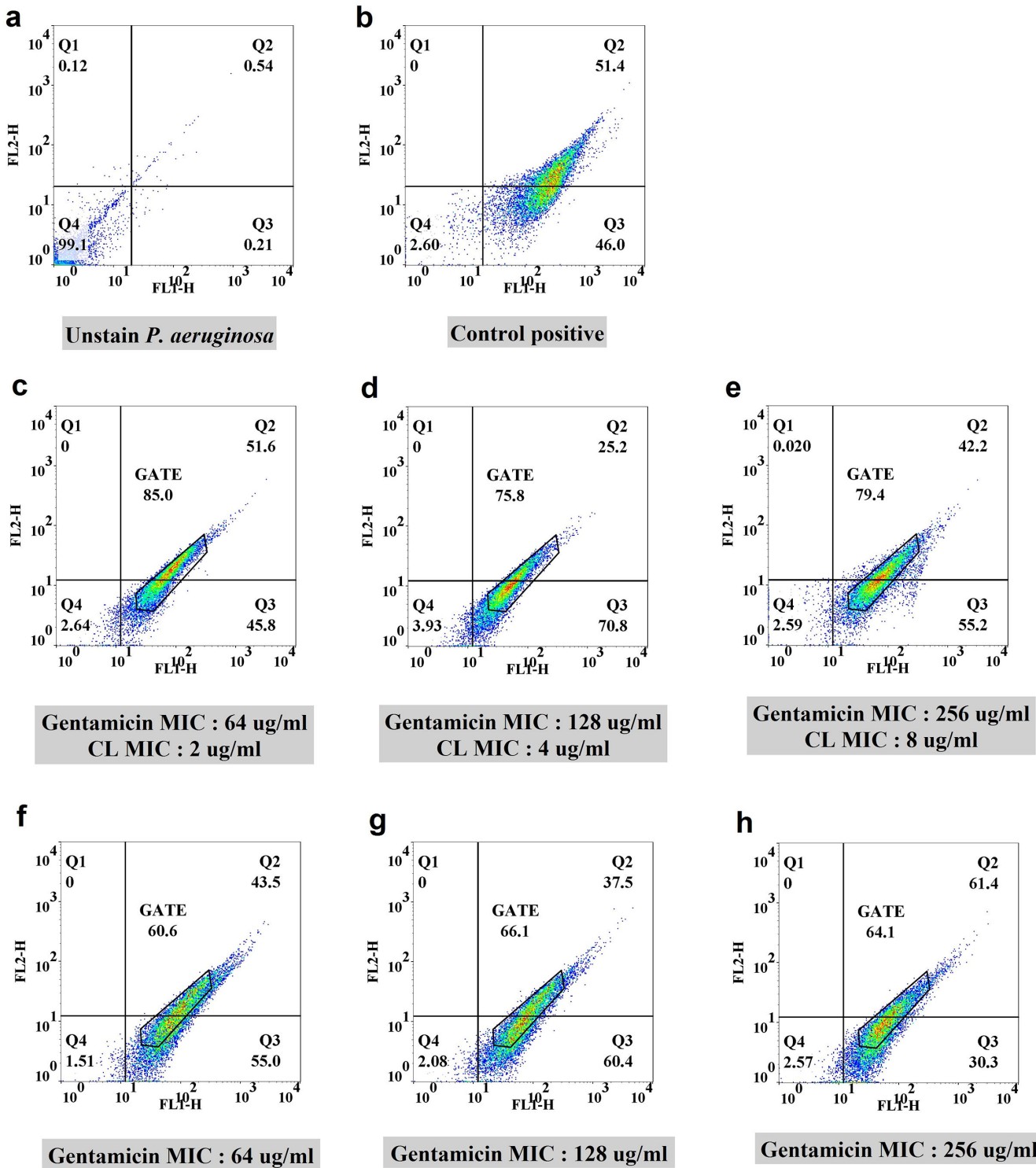

**FIG 3** Flow cytometric analysis of efflux pump inhibition by CL on the selected XDR *P. aeruginosa*. Two-dimensional plots for (a) unstained sample and (b) positive control containing CCCP. (c, d, and e) Sub-inhibitory concentrations (1/8, 1/4, 1/2 MIC) of gentamicin-CL and (f, g, and h) gentamicin at the same concentrations as controls.

intracellular metabolism, increases the permeability of the outer membrane, and disrupts cellular metabolism, ultimately resulting in cell death (19). Li et al. confirmed that the MIC of CGA against *S. aureus* was found to be in the range of 2.5–5.0 mg/mL (20).

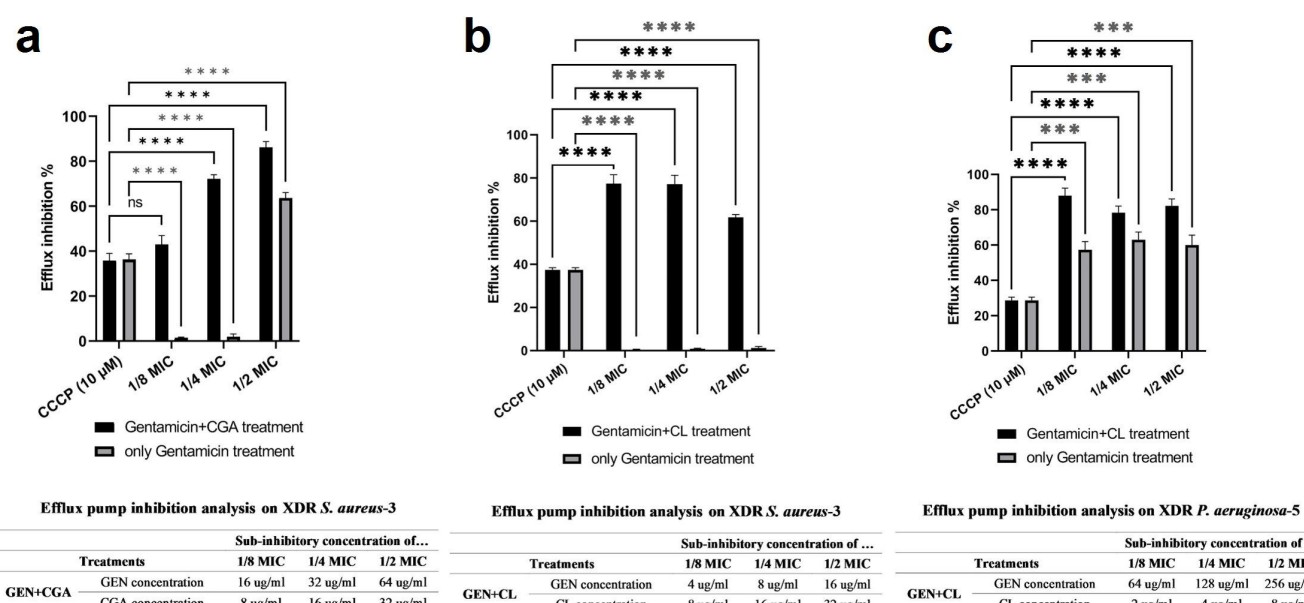

**FIG 4** Efflux pump inhibition analysis. (a, b) Efflux pump inhibition by CGA and CL in combination with gentamicin at the sub-MIC concentrations (1/8, 1/4, and 1/2 MIC) on the selected XDR *S. aureus* and (c) the selected XDR *P. aeruginosa*. The statistical significance of each concentration relative to the corresponding control (CCCP) is summarized with an asterisk (*), with $P > 0.05$, $P \leq 0.05$, $P \leq 0.01$, $P \leq 0.001$, and $P \leq 0.0001$ represented by ns (non-significant), *, **, ***, and ****, respectively.

Furthermore, Su et al. reported that the MIC of CGA against chicken-derived *Staphylococcus saprophyticus* and *P. aeruginosa* P1 was 5 mg/mL (19). According to Lou et al., the CGA MIC against *Bacillus subtilis* was 40 mg/mL (21). These findings support the hypothesis that the structure and function of CGA may contribute to its antimicrobial properties in the following ways: (i) The presence of phenolic hydroxyl groups in the CGA

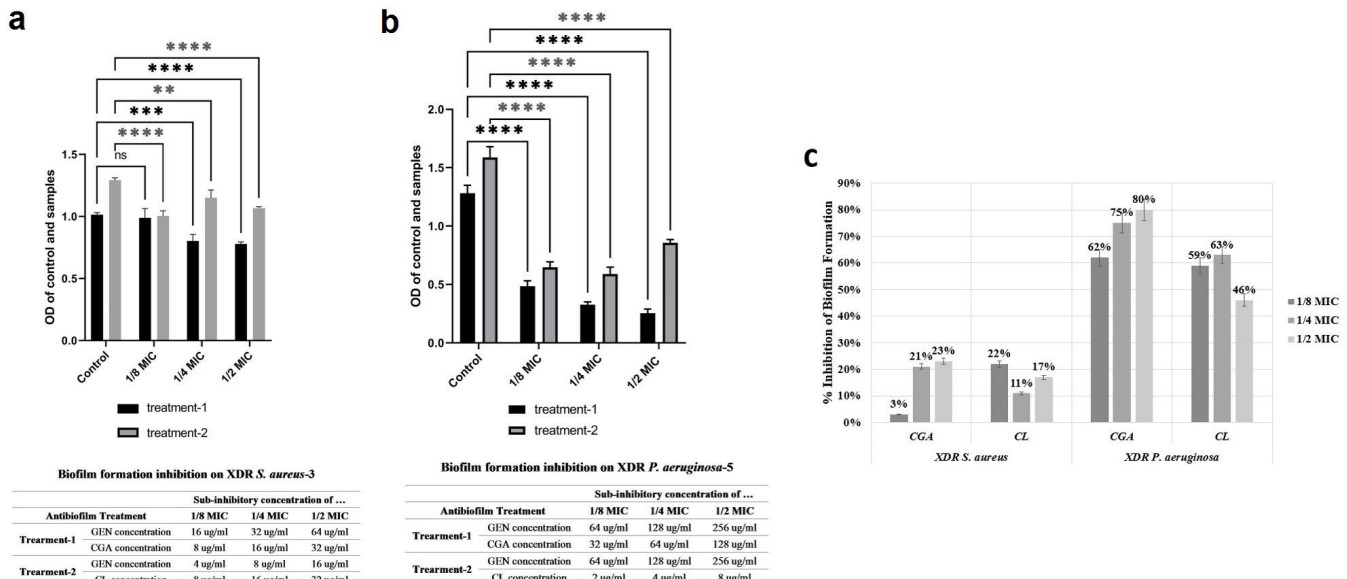

**FIG 5** Antibiofilm assay. The biofilm formation inhibition of CGA and CL in combination with gentamicin at the sub-MIC concentrations (1/8, 1/4, and 1/2 MIC) on the selected (a) XDR *S. aureus* and (b) XDR *P. aeruginosa*. The statistical significance of each concentration relative to the corresponding blank is summarized with an asterisk (*), with $P > 0.05$, $P \leq 0.05$, $P \leq 0.01$, $P \leq 0.001$, and $P \leq 0.0001$ represented by ns (non-significant), *, **, ***, and ****, respectively. (c) The effect of CGA and CL on XDR *S. aureus* and XDR *P. aeruginosa* biofilm, expressed as percentage of inhibition.

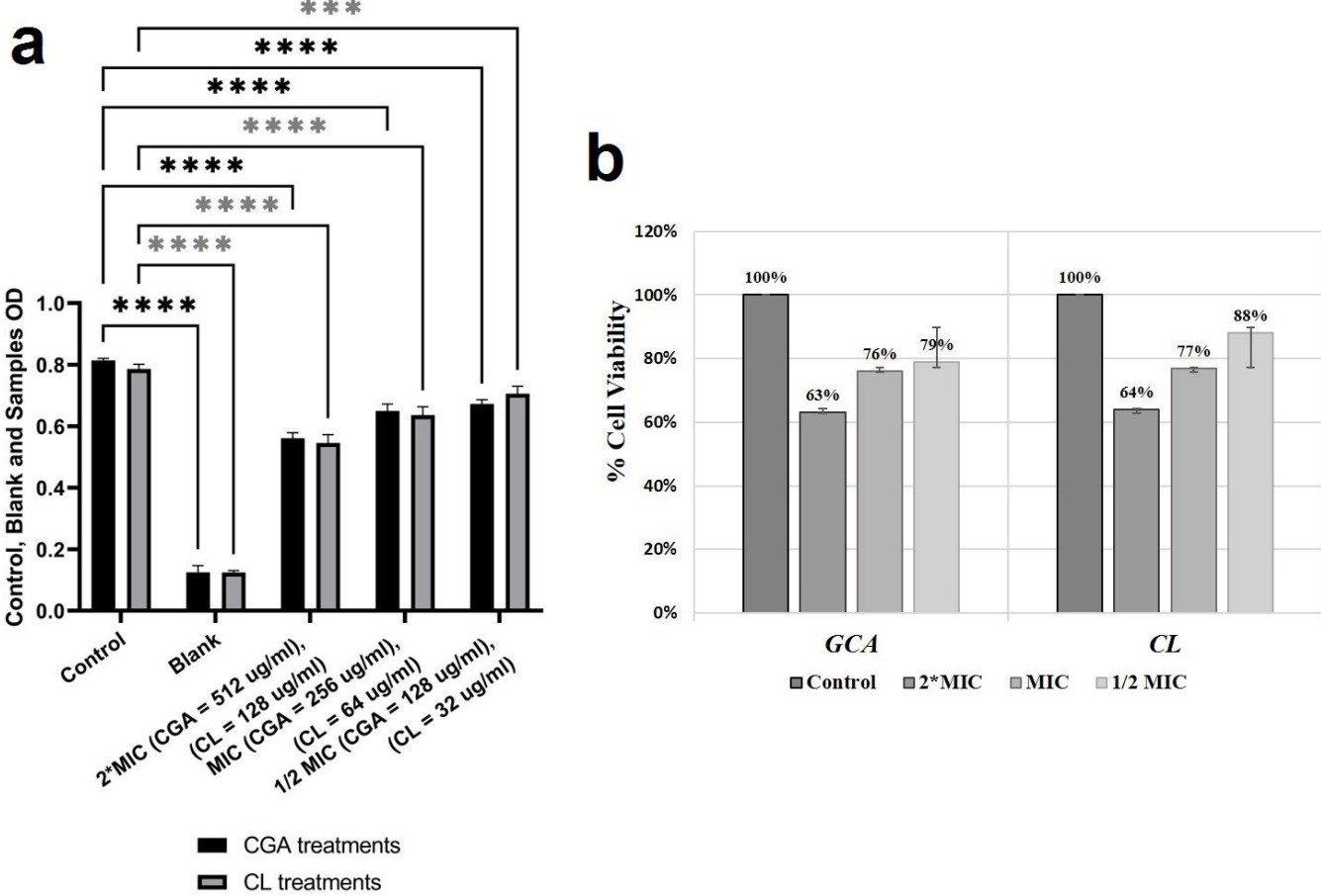

**FIG 6** MTT colorimetric assay. Cell viability investigation on mouse fibroblast NIH/3T3 cell line, by CGA and CL at different concentrations (2× MIC, MIC, 1/2 MIC). (a) The statistical significance of each concentration relative to the corresponding control is summarized with an asterisk (*), with $P > 0.05$, $P \leq 0.05$, $P \leq 0.01$, $P \leq 0.001$, and $P \leq 0.0001$ represented by ns (non-significant), *, **, ***, and ****, respectively. All concentrations had no significant detrimental effect on cell viability. (b) The effect of CGA and CL on NIH/3T3 cell line viability, expressed as percentage of viability.

molecule can influence the activity of relevant metabolic enzymes, leading to decreased metabolism, hindering the metabolic process, and restricting bacterial activity (22). (ii) Due to the high polarity of CGA molecules, they can bind to large molecules, such as lipids present on the bacterial surface, thereby altering the permeability of the bacterial membrane and causing the leakage of cellular contents (21, 23). (iii) CGA can inhibit bacterial aggregation by preventing flagella synthesis and reducing their number (24).

Carnosol and carnosic acid are compounds found in the extract of *Rosmarinus officinalis* L. These compounds have been found to enhance the activity of tetracycline against a strain of *S. aureus* that expresses TetK, increasing its effectiveness by fourfold and twofold, respectively. They also demonstrated a significant reduction in the MIC of erythromycin against a strain of *S. aureus* expressing MsrA, lowering it by eightfold. Moreover, carnosic acid exhibited inhibition of ethidium bromide efflux, which is a substrate for multiple multidrug-resistant pumps, in a strain of *S. aureus* that expresses NorA (25, 26). In addition, the aqueous extracts of *Rosa damascene* and *R. officinalis* were shown to decrease the MICs of several antibiotics against methicillin-resistant *S. aureus* and methicillin-sensitive *S. aureus* strains (25), and this suggests that these extracts may have the potential to enhance the efficacy of antibiotics against these strains. Furthermore, a study conducted by Ghanadian et al. revealed that carnosol had selective antifungal effects against *Candida albicans*, *C. glabrata*, and *C. parapsilosis* (27).

The present study demonstrated that both CGA and CL did not exhibit significant antibacterial activity against XDR strains, as their MIC values were above ≥1,024 µg/mL. However, when combined with gentamicin, CGA showed a 2-log decrease in the MIC of XDR *S. aureus*, while CL showed a more substantial reduction of 4-log in XDR *S. aureus* and 1-log in XDR *P. aeruginosa*. The lower decrease in MIC for XDR *P. aeruginosa* may be attributed to the additional OM found in gram-negative bacteria, which acts as a significant barrier to antibiotic penetration. Gram-negative bacteria possess intrinsic resistance to many antibiotics due to the presence of the OM and potent multidrug efflux pumps. The highly asymmetric OM consists of LPS on the outer leaflet and phospholipids on the inner leaflet, creating a permeability barrier that restricts the entry of antibiotics into the bacterial cell (28).

In our investigation, the modulation of efflux pump activity was assessed at sub-MIC concentrations of 1/8, 1/4, and 1/2. CGA exhibited 39%, 70%, and 19% improvement in efflux pump inhibition in XDR *S. aureus*, while in the case of CL, an approximately 74%, 73.5%, and 62% progression in efflux pump inhibition was observed when comparing the treatment group with the controls. In XDR *P. aeruginosa*, CL demonstrated an improvement in efflux pump inhibition rate by 25%, 10%, and 15%. In flow cytometric investigation, CGA and CL could increase fluorescence relative to the control. This can probably be attributed to the efflux pump inhibition; however, it can also be due to the fluorescence emitted by the compounds. Based on the analyses performed, the excitation and emission wavelengths used to detect CGA and CL were 327–449 nm and 230–330 nm, respectively (29, 30), where these values do not overlap with the 485–538 nm wavelength used to identify Rh 123. It should be noted that while the results suggest a potential potentiating effect of CGA and CL in suppressing bacterial efflux pumps, these findings alone do not definitively conclude that these compounds act as efflux pump inhibitors. Other mechanisms related to efflux pump inhibition, such as membrane destabilization, interference with the proton pump, or other energy sources, need to be considered. Further investigations are required to elucidate the specific mechanisms of action and confirm the efflux pump inhibitory effects of CGA and CL.

Efflux pumps have been found to play multiple roles in the development of biofilms. These include the extrusion of extracellular polymeric substances, quorum-sensing molecules, and quorum-quenching molecules that promote biofilm matrix formation and regulate QS. Efflux pumps can also indirectly influence genes related to biofilm formation and excrete hazardous substances such as metabolic intermediates and antibiotics. They can also affect aggregation by promoting or inhibiting adherence to surfaces and other cells. Biofilm-associated bacteria are known to be more resistant to antibiotics compared to their planktonic counterparts, making traditional antibiotic therapy less effective in eradicating biofilm-embedded bacteria (31).

Studies have shown that efflux pump expression is enhanced in *P. aeruginosa* biofilms. The transcriptomes of developing biofilms and planktonic cultures were compared by Waite et al. and were shown that the expression of the efflux genes was up-regulated when compared to planktonic growth (32). Another study reported that the mexAB-oprM and mexCD-oprJ efflux systems were necessary for the production of biofilm in the presence of azithromycin, and the MexCD-OprJ efflux system was up-regulated in *P. aeruginosa* biofilm (33). In the case of *S. aureus*, MFS-type efflux pumps have been implicated in biofilm formation, but the behavior of most *S. aureus* efflux genes in biofilms is still unknown. Glucose fermentation by *S. aureus* during biofilm development leads to the accumulation of organic acids, which can limit biofilm growth and activate stress response pathways. Efflux pumps, such as NorB, may play a role in protecting biofilms from the damaging effects of organic acids (34, 35). Overall, further research is needed to explore the involvement of efflux pumps from different superfamilies and their specific contributions to biofilm formation in various bacterial species (31).

In our study, the effect of CGA and CL as potential efflux pump inhibitors on biofilm formation was investigated using sub-MIC concentrations of gentamicin and the compounds. The results showed that both CGA and CL exhibited inhibitory effects

on biofilm formation in XDR *S. aureus* and XDR *P. aeruginosa*, with stronger inhibition observed in XDR *P. aeruginosa*. CGA showed a range of inhibition percentages in XDR *S. aureus* and XDR *P. aeruginosa*, with higher percentages observed at higher concentrations. Similarly, CL showed varying degrees of suppression in biofilm formation, with the highest inhibition at the highest concentration tested. Previous studies have also reported the effects of compounds such as carnosol on bacterial motility and biofilm formation in *S. aureus*, indicating their potential role in modulating biofilm development (36). In *P. aeruginosa*, QS is known to be essential for biofilm growth, and efflux pumps can play a significant role in the transport of important substances, including acyl-homoserine lactones (AHLs), which are required for biofilm formation. On the other hand, gram-positive bacteria, such as *S. aureus*, do not produce AHLs in their biofilms. Therefore, the greater inhibition of biofilm formation in XDR *P. aeruginosa* compared to XDR *S. aureus* could be attributed to the targeting of AHLs in biofilm development (37). Overall, the results suggest that CGA and CL may have potential as inhibitors of biofilm formation, with stronger effects observed in XDR *P. aeruginosa*, which may be likely due to the specific mechanisms and pathways involved in biofilm development in different bacterial species.

The cytotoxicity results of CGA and CL at a concentration of 2× MIC showing feeble cell death suggest that these compounds may have a low level of toxicity and could potentially be considered for clinical applications.

## MATERIALS AND METHODS

### Chemicals and reagents

The materials and reagents used in the study, including chlorogenic acid as a phenolic compound and dimethyl sulfoxide (DMSO), were acquired from Merck (Germany). Gentamicin, 3-(4,5-dimethylthiazol-2-yl)-2,5-diphenyltetrazolium bromide (MTT), and carbonyl cyanide-m-chlorophenylhydrazone were purchased from Sigma (USA). Fibroblast NIH/3T3 cell line was obtained from Pasteur Institute, Tehran, Iran. The bacteria *S. aureus* (ATCC:25923) and *P. aeruginosa* (ATCC:27853) were obtained from the Iranian Biological Resource Center, Tehran, Iran. Rhodamine-123 (Rh123) was supplied from Thermo Fisher (USA), and phosphate-buffered saline (PBS), fetal bovine serum (FBS), and RPMI were supplied from Bio-Idea Co (Iran). Muller Hinton agar (MHA), Muller Hinton broth (MHB), and brain-heart infusion (BHI) broth were purchased from Quelab (Montreal, Canada). Carnosol as a diterpenoid compound was isolated and identified from *Salvia abrotanoides* by Ghanadian et al. (27). These materials and reagents were used for various experiments and assays conducted in the study to evaluate the antimicrobial, efflux pump inhibitory, and antibiofilm activities of CGA and CL.

### Bacterial isolation

Clinical strains of *S. aureus* and *P. aeruginosa* were obtained from two medical institutions: Al-Zahra referral hospital and Imam Musa Al-Kazim Burn Center, and from patients with wound infection and burns. Samples were obtained from wounds by utilizing methods such as biopsy, wound fluid aspiration, and wound swab, as necessary for each patient. The confirmed strains were stored in BHI broth supplemented with 20% glycerol at −20°C until further analysis.

### Antibiotic susceptibility testing

After preparation of fresh overnight culture, Kirby-Bauer disk diffusion was used to determine the antibiotics susceptibility of bacterial strains. The sensitivity of *S. aureus* strains to ciprofloxacin (5 µg), gentamicin (10 µg), clindamycin (2 µg), doxycycline (30 µg), penicillin (10 µg), erythromycin (15 µg), trimethoprim/sulfamethoxazole (5 µg), cefoxitin (30 µg), and chloramphenicol (30 µg) discs and *P. aeruginosa* strains to ceftazidime (30 µg), cefepime (30 µg), piperacillin/tazobactam (100/10 µg), gentamicin

(10 µg), meropenem (10 µg), and ciprofloxacin (5 µg) (purchased from Padtan Teb, Tehran, Iran) after 18–24 h of incubation was investigated on MHA medium based on Clinical and Laboratory Standards Institute standards individually (17).

## Determination of minimum inhibitory concentration

In this study, the MICs of gentamicin, CCCP, CGA, and CL were investigated for all XDR strains of *S. aureus* and *P. aeruginosa*. The MICs were determined using the broth microdilution method in 96-well plates following the guidelines provided by reference (17) as well as the manufacturer's instructions (17). Serial twofold dilutions of antibiotic (such as gentamicin) and compounds (such as CCCP, CGA, or CL) were prepared in triplicate, individually, and the dilutions ranged from the highest concentration of 1,024 µg/mL to the lowest concentration of 2 µg/mL (1,024, 512, 256, 128, 64, 32, 16, 8, 4, and 2 µg/mL). After preparing the dilutions, a standard bacterial suspension containing $1$–$5 \times 10^5$ colony-forming units (CFU)/mL was added to each well. After 18–24 h of incubation at 37℃, the lowest concentration of drugs that inhibited the visible growth of bacteria was defined as MIC. The performance of the assay was verified using reference strains of *S. aureus* (ATCC 25923) and *P. aeruginosa* (ATCC 27853).

## Assessment of antimicrobial activity alteration by checkerboard assay

To evaluate the potential of phenolic and diterpene compounds as microbial resistance modulators, the MIC was determined with gentamicin, in the presence of the compounds at different concentrations (1/8 MIC to 2× MIC), compared to the negative control DMSO and positive control containing microorganisms. Suspensions of $1$–$5 \times 10^5$ CFU/mL of bacterial culture in the BHI medium were prepared and distributed into microtiter trays for all XDR strains. The microplates were incubated at 37℃ for 24 h and then bacterial growth evaluated. To assess the effects of drugs combination, fractional inhibitory concentration (FIC) indices were calculated as $FIC^A + FIC^B$, where $FIC^A$ and $FIC^B$ represent the minimum concentrations inhibiting bacterial growth for drugs A and B, respectively. $FIC^A = MIC^A$ combination/$MIC^A$ alone and $FIC^B = MIC^B$ combination/$MIC^B$ alone. The FIC index was calculated based on the following (equation 1):

$$\sum FICI = FIC(A) + FIC(B)$$

The interpretation of the FIC index is as follows: synergistic (≤0.5), additive (0.5–1.0), indifferent (>1.0), or antagonistic (>4.0) (38).

## Efflux pump screening by CCCP

To examine efflux pump activity, CCCP at a final concentration of 25 µg/mL was added to each microtiter plate well containing 2–1,024 µg/mL of gentamicin. A stock solution of CCCP was prepared previously at 5 mg/mL in DMSO. The final concentration of CCCP in the MHB was 10 mg/L with a DMSO concentration of 0.2% (39). Once more, the MIC of gentamicin was determined after treatment with CCCP. As an energy-dependent decoupler that inhibits antibiotic efflux, the CCCP molecule aids in the understanding of the efflux mechanism. A fourfold reduction in MIC following CCCP application was considered indicative of significant efflux pump activity (40). All the experiments were performed in triplicates, and the antibiotic-free well containing CCCP served as control.

## Evaluation of dye accumulation by flow cytometry

Efflux assays were conducted to assess the ability of CGA and CL to promote the accumulation of Rhodamine-123 (Rh123) in XDR strains. Rh123 is commonly used as a tracer dye for membrane transport (41). For the efflux experiments, sub-MIC concentrations of CGA and CL (1/8, 1/4, and 1/2 MIC) were used. The most resistant strain of XDR *S. aureus* and XDR *P. aeruginosa* was cultured overnight in 10 mL of BHI broth in a shaking incubator at 37℃. Two sets of cultures were prepared, such as one with CCCP as

a reference inhibitor (positive control) and another without CCCP (negative control). The bacterial strains were diluted in BHI broth until a final optical density (OD) 600 of 0.08–0.13 was achieved. The bacterial suspension was then centrifuged at 3,000 RPM for 5 minutes, and the supernatant was carefully removed. The pellet was washed three times with 10 mL of PBS to remove any residual media or debris. After the final washing step, the pellet was resuspended in PBS using a vortex mixer. The bacterial cultures and test components were then combined in microtubes as working solutions. Blank solutions containing test substrates were prepared and mixed with PBS without bacterial cultures. Rh123 was prepared at a final concentration of 10 µM, added to all flow cytometry tubes, which were then wrapped in aluminum foil to protect them from light exposure. The tubes were incubated in the dark at 37°C for 30 minutes. Fluorescence was measured with flow cytometer FACSCalibur (BD Biosciences, San Jose, CA, USA), with excitation and emission wavelengths of 485 and 538 nm (FL-2 channel), respectively (42). Analyses were performed with the acquisition of at least 10,000 events per sample.

## Biofilm formation inhibition assay

The assay was conducted in 96-well microtiter plates following the protocol described by Christensen et al., with some modifications (43). The bacterial suspension (1–5 × $10^5$ CFU/mL of XDR *S. aureus* and XDR *P. aeruginosa*), CGA, and CL at concentrations less than MIC (1/8, 1/4, and 1/2 MIC) were added to each well. Then the microplates were incubated at 37°C for 48 h under aerobic conditions. After the incubation period, planktonic cells were removed by washing the wells twice with distilled water. The biofilms were stained with only 150 µL of crystal violet and allowed to incubate for 15 minutes. Subsequently, biofilms were fixed by adding 150 µL of ethanol 96% for an additional 15 minutes. The wells were then washed twice with water to remove the excess crystal violet. To quantify the biofilm formation, 150 µL of 33% acetic acid was added to each well. The microplates were read using a BIO-RAD Microplate Reader (Model 680) at a wavelength of 570 nm. The study was performed in triplicates. Blank wells containing sterile MHB were used as the negative control, and the results were reported as the mean ± SD. The percentage of biofilm formation inhibition was calculated using the following equation (equation 2) (44, 45):

$$\%\text{Biofilm formation inhibition} = \frac{\text{OD blank} - \text{OD sample}}{\text{OD blank}} \times 100$$

## MTT assay for cell viability analysis

The cytotoxicity of CGA and CL against mouse fibroblast cell line was evaluated by a colorimetric assay using MTT as described by Tim Mosmann with some modifications (46). The NIH/3T3 cell line stock cells were cultured in RPMI-1640 as an appropriate complete cell culture medium at 37°C and 5% $CO_2$ air humidified in a 96-well cell culture microplate. Each well was filled with 100 µL of culture media. Cells were treated with CGA and CL previously diluted in DMSO and culture medium (a final DMSO concentration of less than 2% in the well) for each compound in the first well of each row. Then, a serial dilution of the compounds (ranging from 2× MIC, MIC, and 1/2 MIC) was prepared for the remaining wells. Prior to adding the cells, the cells were pre-counted in a Neubauer chamber, and cell viability was confirmed using the Trypan blue dye. Subsequently, 100 µL of a cell suspension at a concentration of 1 × $10^4$ cells per well was dispensed into the microplate wells. A control well containing only cells with culture medium and a blank well containing only culture medium were also included. The microplate was then incubated for 24 h. After the incubation period, the wells were washed three times with culture medium containing FBS to remove any residual compounds or substances, and subsequently, 100 µL of MTT solution (working solution: 5 mg/mL was diluted 10 times in medium) added in the dark, and the plate was incubated for an additional 2–3 h. Finally, the ability of cells to sequester and reduce MTT to form formazan crystals was assessed. The crystals were diluted with a DMSO solution (100 µL/well), and the plate was gently

stirred for 30 minutes to ensure complete dissolution of the crystals. Subsequently, the absorbance was measured in a BIO-RAD Microplate Reader at 570 nm. The assay was done in triplicate, and results were reported as mean ± SD. The cell viability was calculated as a percentage relative to the control using the following formula (equation 3) (47):

$$\%\text{Cell viability} = \frac{\text{Sample Absorbance} - \text{Blank Absorbance}}{\text{Cell growth control Absorbance} - \text{Blank Absorbance}} \times 100$$

## Statistical analysis

The assays were performed in triplicate, and the results were presented as the mean ± SD. Statistical analysis was conducted using two-way ANOVA followed by Tukey's post hoc test using GraphPad Prism 9.5.1 software. Statistical significance was considered when the $P$-value was less than 0.05 ($P < 0.05$).

## Conclusion

In summary, CGA and CL demonstrated promising potentiating antimicrobial effects against XDR strains of *S. aureus* and *P. aeruginosa*, suggesting their probably potential as candidates for addressing nosocomial pathogens. They exhibited significant suppression of efflux pump activity, indicating a possible successful inhibition of this mechanism. Moreover, all substances effectively inhibited biofilm formation while showing minimal cytotoxicity. However, further advancement to clinical trials is needed to evaluate the feasibility of utilizing CGA and CL for reversing bacterial XDR efflux and determining their efficacy against biofilms. These trials will provide valuable insights into the practical applications of these compounds in combating drug-resistant infections.

## ACKNOWLEDGMENTS

This work was funded by Isfahan University of Medical Sciences, Isfahan, Iran, through Grant No. 340106.

H.F. played a significant role in coordinating and overseeing the entire project and was responsible for securing the funding necessary for its execution. Meanwhile, the experiments were carried out by M.S., who was also responsible for drafting the manuscript. The identification and isolation of carnosol from *Salvia abrotanoides* were performed by M.G. Additionally, H.F. and V.K. were primarily involved in designing the experiments and analyzing the data collected.

## AUTHOR AFFILIATIONS

[1]Department of Bacteriology and Virology, School of Medicine, Isfahan University of Medical Sciences, Isfahan, Iran
[2]Department of Pharmacognosy, Pharmaceutical Sciences Research Center, School of Pharmacy and Pharmaceutical Sciences, Isfahan University of Medical Sciences, Isfahan, Iran

## AUTHOR ORCIDs

Hossein Fazeli http://orcid.org/0000-0003-4054-8671

## FUNDING

| Funder | Grant(s) | Author(s) |
| --- | --- | --- |
| Isfahan University of Medical Sciences (IUMS) | 340106 | Hossein Fazeli |

## DATA AVAILABILITY

Upon receipt of a reasonable request, the corresponding author will provide accessibility to the data sets that were generated and/or analyzed during the course of the current inquiry.

## ETHICS APPROVAL

The study was approved by the Ethical Committee of Isfahan University of Medical Sciences (approval number: IR.MUI.MED.REC.1401.050). Every experiment was carried out in accordance with the applicable rules and regulations.

## ADDITIONAL FILES

The following material is available online.

### Open Peer Review

**PEER REVIEW HISTORY (review-history.pdf).** An accounting of the reviewer comments and feedback.

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
