## [Reviewer comments · Microbiology Spectrum]

Microbiology Spectrum

Evaluation of Chlorogenic Acid and Carnosol for Anti-Efflux Pump and Anti-Biofilm Activities against Extensively Drug-Resistant (XDR) Strains of *Staphylococcus aureus* and *Pseudomonas aeruginosa*

mohaddeseh sheikhy, Vajihe Karbasizade, Mustafa Ghanadian, and Hossein fazeli

Corresponding Author(s): Hossein fazeli, Isfahan University of Medical Sciences

Review Timeline:

Submission Date:	November 14, 2023
Editorial Decision:	January 22, 2024
Revision Received:	April 14, 2024
Accepted:	April 18, 2024

Editor: Krisztina Papp-Wallace

Reviewer(s): Disclosure of reviewer identity is with reference to reviewer comments included in decision letter(s). The following individuals involved in review of your submission have agreed to reveal their identity: sreekanth reddy basireddy (Reviewer #1); Amjed alsultan (Reviewer #2)

Transaction Report:

DOI: <https://doi.org/10.1128/spectrum.03934-23>

Re: Spectrum03934-23 (Evaluation of Chlorogenic Acid and Carnosol for Anti-Efflux Pump and Anti-Biofilm Activities against Clinical Extensively Drug-Resistant (XDR) Strains of *Staphylococcus aureus* and *Pseudomonas aeruginosa*)

Dear Dr. Hosein Fazeli:

Thank you for the privilege of reviewing your work. Below you will find my comments, instructions from the Spectrum editorial office, and the reviewer comments.

Revision Guidelines

Sincerely,
Krisztina Papp-Wallace
Editor
Microbiology Spectrum

Reviewer #1 (Comments for the Author):

In this study, the authors tried to evaluate the ant-efflux and antibiofilm activity of chlorogenic acid and carnosol, two plant derived compounds, against drug-resistant *S. aureus* and *P. aeruginosa* clinical strains. A series of experiments were conducted in selected strains of XDR *S. aureus* and *P. aeruginosa* which include antibiotic potentiating effects, efflux pump inhibition, biofilm inhibition and cell viability testing to identify the activity of these two phytochemicals in combination with gentamicin.

The study could have been improved by testing a greater number of resistant isolates instead of testing single selected isolates, at least for checkerboard assay and biofilm inhibition assays, which are easy to perform. Confidently attributing the results of a single strain to all clinical isolates is challenging, given the variations in resistance mechanisms observed among different clinical isolates. Though we could observe some antibiotic potentiating effect in the 2 selected isolates tested, even when combined with carnosol and chlorogenic acid, the gentamicin's MIC remains significantly higher than the cutoff value. Similarly, minimal inhibition in biofilm formation was observed for *S. aureus* which may also could vary between different strains.

Despite the well-done study, the author may need to address the following concerns.

1. Line 16 - It is mentioned that, a total of 12 XDR strains were studied. Certain strains seem to be MDR pathogens rather than XDR ones, as they are sensitive to many drugs. Only disc diffusion method is used to identify the sensitivity of antibiotics and all the antibiotics active against *S. aureus* and *P. aeruginosa* were not tested/shown (Table 1 and 2). This needs clarification. Also, Gentamicin is not used as a monotherapy and is not the treatment of choice for XDR *S. aureus*.

2. Line 103- It is mentioned that MIC values of chlorogenic acid and carnosol are equal/above 1024 µg/ml. Are MIC values assessed for all clinical isolates, or is the testing limited to specific individual isolates of *S. aureus* and *P. aeruginosa* for chlorogenic acid and carnosol? Similarly, in efflux pump inhibition assay and biofilm inhibition assay it is mentioned that 1/2 MIC values for *S. aureus* -3 and *P. aeruginosa* -5 strain as 64 µg/ml and 256 µg/ml respectively (figure 4 and 5), whereas actual MIC for these strains is 512 µg/ml and 1024 µg/ml as shown in Table 1 and 2. Discrepancies should be corrected.

3. Line 106- CGA had an additive effect on *P. aeruginosa* strain. In table 3 it is mentioned indifference. Discrepancy should be corrected.

4. Line 165 - for both XDR strains which selected was - should be rephrased

Reviewer #2 (Comments for the Author):

the manuscript are needed to improve in scientific level. All part should be revised and rearranged.

Reviewer comments

In this study, the authors tried to evaluate the ant-efflux and antibiofilm activity of chlorogenic acid and carnosol, two plant derived compounds, against drug-resistant *S. aureus* and *P. aeruginosa* clinical strains. A series of experiments were conducted in selected strains of XDR *S. aureus* and *P. aeruginosa* which include antibiotic potentiating effects, efflux pump inhibition, biofilm inhibition and cell viability testing to identify the activity of these two phytochemicals in combination with gentamicin.

The study could have been improved by testing a greater number of resistant isolates instead of testing single selected isolates, at least for checkerboard assay and biofilm inhibition assays, which are easy to perform. Confidently attributing the results of a single strain to all clinical isolates is challenging, given the variations in resistance mechanisms observed among different clinical isolates. Though we could observe some antibiotic potentiating effect in the 2 selected isolates tested, even when combined with carnosol and chlorogenic acid, the gentamicin's MIC remains significantly higher than the cutoff value. Similarly, minimal inhibition in biofilm formation was observed for *S. aureus* which may also could vary between different strains.

Despite the well-done study, the author may need to address the following concerns.

1. Line 16 – It is mentioned that, a total of 12 XDR strains were studied. Certain strains seem to be MDR pathogens rather than XDR ones, as they are sensitive to many drugs. Only disc diffusion method is used to identify the sensitivity of antibiotics and all the antibiotics active against *S. aureus* and *P. aeruginosa* were not tested/shown (Table 1 and 2). This needs clarification. Also, Gentamicin is not used as a monotherapy and is not the treatment of choice for XDR *S. aureus*.
2. Line 103- It is mentioned that MIC values of chlorogenic acid and carnosol are equal/above 1024 µg/ml. Are MIC values assessed for all clinical isolates, or is the testing limited to specific individual isolates of *S. aureus* and *P. aeruginosa* for chlorogenic acid and carnosol? Similarly, in efflux pump inhibition assay and biofilm inhibition assay it is mentioned that ½ MIC values for *S. aureus* -3 and *P. aeruginosa* -5 strain as 64 µg/ml and 256 µg/ml respectively (figure 4 and 5), whereas actual MIC for these strains is 512 µg/ml and 1024 µg/ml as shown in Table 1 and 2. Discrepancies should be corrected.
3. Line 106- CGA had an additive effect on *P. aeruginosa* strain. In table 3 it is mentioned indifference. Discrepancy should be corrected.
4. Line 165 – for both XDR strains which selected was – should be rephrased

Evaluation of Chlorogenic Acid and Carnosol for Anti-Efflux Pump and Anti-Biofilm Activities against Clinical Extensively Drug-Resistant (XDR) Strains of *Staphylococcus aureus* and *Pseudomonas aeruginosa*

The manuscript titled " Evaluation of Chlorogenic Acid and Carnosol for Anti-Efflux Pump and Anti-Biofilm Activities against Clinical Extensively Drug-Resistant (XDR) Strains of *Staphylococcus aureus* and *Pseudomonas aeruginosa*" presents an interesting and relevant study in the field of microbiology and antibiotic resistance.

The study design, contents and information of the manuscript are needed to improve in scientific level. All part should be revised and rearranged.

Following requests are necessary to be addressed to improve quality of the study.

Title:

- " clinically" bit confuse, the tested bacterial strain is in vitro resist to wide range of antibiotic not in vivo or resist to antimicrobial therapy during infection.

Abstract:

- Line 13, need to explain on which based you selected these two compounds (previous study or documented action on the other bacteria rather than staphylococcus and pseudomonas).
- Line, did you detect the MIC of CCCP against staph. And pseudomonas
- Line 18, why you choose gentamycin, many different antimicrobial agent is more important as anti-staph. Infection. You need to show that study deal with staphylococcus strains that cause skin infection
- In this section you need to suggest a possible mechanism for inhibition and is this compound inhibit specific type of efflux pump protein or not.

Introduction:

- In this section you need to paragraph described the family of efflux pump system in staphylococcus and pseudomonas and those that efflux gentamycin.
- Line 57, if you use expression profile (with qPCR or NGS) for selected efflux protein it will good proof for overexpression on down expression of the efflux pump that results from the tested efflux inhibitor
- Line 67, need to explain role of efflux pump in virulence and quorum sensing in bacteria to show how EPIs are important

Results:

- Line 90, you need to mention details about isolated strain, is it from skin or other part, type of infection, host (human or animal).

- Line 102, basically all EPIs has antimicrobial activity for example Reserpine which well documented as A potent EPIs. Just double check your statement that chromogenic acid and carnosol have no antibacterial activity.
- Line 112, you need to detect the MIC of CCCP also explain why use this inhibitor with gentamicin.
- Line 123, need to mention some details about isolated strain (host, type of infection etc.)
- Line 125, you need reference " CGA, and CL that did 124 not impact cell viability "
- Line139, *P. aeruginosa* this isolated associated with human infection or not did host receive antibiotic (please provide information about isolates)
- Line 162, at least mention that you use cell to check viability with some details

Discussion:

In this section you need to

- Explain why use two assumed EPIs
- Relationship between biofilm and Efflux pump and how EPIs are important
- Suggested the source of EPIs (plant source or active chemical agent)
- Suggest possible mechanisms for both RPIs that used
- Is the tested EPIs have selected action or not?
- Explain limitation of the study, didn't do expression profile of efflux pump to prof inhibition action of EPIs, used on type of antibiotic

Methods:

- Line 305, " from patients with 304 wound infection and burns" need to show how isolated the bacteria
- Line 308, need to mention stage of bacterial growth that used in sensitivity assay.
- Line 318, it is perfect to used agar based method to confirm MIC
-

Finally, all above, the manuscript is minor revision

Here is a point-by-point response to the reviewers' comments and concerns:

Comments from Reviewer 1

- **Comment 1:** *Line 16 – It is mentioned that, a total of 12 XDR strains were studied. Certain strains seem to be MDR pathogens rather than XDR ones, as they are sensitive to many drugs. Only disc diffusion method is used to identify the sensitivity of antibiotics and all the antibiotics active against *S. aureus* and *P. aeruginosa* were not tested/shown (Table 1 and 2). This needs clarification. Also, Gentamicin is not used as a monotherapy and is not the treatment of choice for XDR *S. aureus*.*

Response: Thanks for your kind reminders. In response to your guidance, amendments were made to the manuscript on page 1/lines 14-20. Furthermore, the use of the disc diffusion method for antibiotic sensitivity testing is a recognized and validated approach in microbiology. As you know, extensively drug resistant (XDR) was defined as nonsusceptibility to at least one agent in all but two or fewer antimicrobial categories (i.e., bacterial isolates remain susceptible to only one or two antimicrobial categories). However, we ideally believe that a broader array of antibiotic discs would have been preferable, but, readily accessible antibiotic discs were employed due to financial constraints limiting the ability to expand the disc selection. Despite the infrequent use of gentamicin as monotherapy for treating XDR *S. aureus* infections, it is commonly employed in the management of post-burn infections caused by this pathogen. In addition, one of the purposes of this study is to investigate the combined effect of chlorogenic acid and carnosol in association with gentamicin.

- **Comment 2:** *Line 103- It is mentioned that MIC values of chlorogenic acid and carnosol are equal/above 1024µg/ml. Are MIC values assessed for all clinical isolates, or is the testing limited to specific individual isolates of *S. aureus* and *P. aeruginosa* for chlorogenic acid and carnosol? Similarly, in efflux pump inhibition assay and biofilm*

inhibition assay it is mentioned that $\frac{1}{2}$ MIC values for *S. aureus* -3 and *P. aeruginosa* -5 strain as 64 $\mu\text{g/ml}$ and 256 $\mu\text{g/ml}$ respectively (figure 4 and 5), whereas actual MIC for these strains is 512 $\mu\text{g/ml}$ and 1024 $\mu\text{g/ml}$ as shown in Table 1 and 2. Discrepancies should be corrected.

- Response:** Thanks for your question. In response to your query, yes, the MIC values of chlorogenic acid and carnosol were assessed for all the clinical isolates under investigation. And as mentioned in the text of the manuscript, “Gentamicin MIC value was also determined for all clinical strain using microplate broth microdilution method. Clinical strains had different gentamicin MIC values that ranged from 8 to 512 for *S. aureus* and 2 to 1024 $\mu\text{g/ml}$ for *P. aeruginosa*, respectively.” Page 3/lines 97-99. It is also mentioned in the rest of this section that “The most resistant XDR strain of each bacterium was used for subsequent steps (*S. aureus*-3, and *P. aeruginosa*-5).” Page 4/lines 101-102. The actual gentamicin MIC for *S. aureus*-3, and *P. aeruginosa*-5 are 512 $\mu\text{g/ml}$ and 1024 $\mu\text{g/ml}$, respectively, as mentioned by the respected reviewer above. Finally, the results obtained from the checkerboard assay, as described thoroughly on page 4/lines 103-113, were utilized in the efflux pump inhibition assay and inhibition of biofilm formation. For further clarification, Table 3 of the manuscript can be consulted, as it is also referenced below.

Table 3: Assessment of gentamicin antimicrobial activity in combination with CGA and CL by checkerboard assay.

Bacteria	Minimum Inhibitory Concentrations (MICs) of ...							Checkerboard Results based on	
	GEN and compounds before treatment			GEN and compounds after treatment				interpretation of FIC index	
	GEN	CGA	CL	GEN+CGA	GEN+CL	CGA	CL	GEN+CGA	GEN+CL
XDR S. aureus -3	512 $\mu\text{g/ml}$	1024 $\mu\text{g/ml}$	1024 $\mu\text{g/ml}$	128 $\mu\text{g/ml}$	32 $\mu\text{g/ml}$	64 $\mu\text{g/ml}$	64 $\mu\text{g/ml}$	Synergism (0.312)	Synergism (0.125)
XDR P. aeruginosa -5	1024 $\mu\text{g/ml}$	1024 $\mu\text{g/ml}$	1024 $\mu\text{g/ml}$	512 $\mu\text{g/ml}$	512 $\mu\text{g/ml}$	256 $\mu\text{g/ml}$	16 $\mu\text{g/ml}$	Indifference (0.750)	Synergism (~ 0.5)

GEN: Gentamicin, CGA: Chlorogenic acid, CL: Carnosol, FIC: Fractional Inhibitory Concentration, XDR: Extensively Drug Resistant.

To investigate the efflux pump activity of XDR *S. aureus*, subinhibitory concentrations (1/8, 1/4 and 1/2 MIC) of gentamicin in combination with sub-MIC concentrations of CGA (16, 32, and 64 $\mu\text{g/ml}$ along with 8, 16, and 32 $\mu\text{g/ml}$) and CL (4, 8, and 16 $\mu\text{g/ml}$ along with 8, 16, and 32 $\mu\text{g/ml}$) were evaluated, respectively. Since CGA did not exhibit a synergistic effect on XDR *P. aeruginosa*, investigation of efflux pump activity was conducted using subinhibitory concentrations (1/8, 1/4 and 1/2 MIC) of gentamicin (64,

128, and 256 µg/ml) in combination with sub-MIC concentrations of CL (2, 4, and 8 µg/ml).

- **Comment 3:** Line 106- CGA had an additive effect on *P. aeruginosa* strain. In table 3 it is mentioned indifference. Discrepancy should be corrected.
- **Response:** I appreciate the reminder. According to the Fractional inhibitory concentration (FIC) index interpretation, an FICI between 0.50 and 1.00 represents an additive effect. In this case, CGA had an additive effect on the selected XDR *P. aeruginosa*-5 strain (FICI = 0.75). Table 3 was revised accordingly.
- **Comment 4:** Line 165 – for both XDR strains which selected was – should be rephrased
- **Response:** Thank you for your comment. In response to your suggestion, the phrase “for both XDR stains which selected was” revised to “the maximum MIC values for CGA and CL in combination with gentamicin were 64 µg/ml for XDR strains of *S. aureus*-3 and *P. aeruginosa*-5” on page 5/lines 169-170.

Comments from Reviewer 2

Title:

- **Comment 1:** "clinically" bit confuse, the tested bacterial strain is in vitro resist to wide range of antibiotic not in vivo or resist to antimicrobial therapy during infection.
- **Response:** Thank you for pointing this out. The term "clinical" was used to emphasize that the strains investigated in this study were isolated from nosocomial infections, as well as from patients with wound and burn infections that were resistant to antibiotic treatment. The title has been revised in accordance with your suggestion (Page 1/ line 2).

Abstract:

- **Comment 1:** *Line 13, need to explain on which based you selected these two compounds (previous study or documented action on the other bacteria rather than staphylococcus and pseudomonas).*
- **Response:** Thanks for your comment. However, we believe that addressing the importance of these compounds, which somehow influenced the selection and study of their potential in this study would be more appropriate in the introduction section because the word limit in the abstract section does not allow that this issue should be addressed well. A background regarding this matter was outlined on page 3/lines 77-84.
- **Comment 2:** *Line, did you detect the MIC of CCCP against staph. And pseudomonas*
- **Response:** Thank you for your question. In response, a dilutions series of CCCP were prepared ranging from a concentration of 2-1024 µg/ml using MIC microdilution broth before conducting the efflux pump screening. The growth of *S. aureus* and *P. aeruginosa* strains was examined in this series of dilutions. It was observed that the growth of both *S. aureus* and *P. aeruginosa* strains remained unaffected until the concentration of 1024 µg/ml that was investigated. Finally, to examine efflux pump activity, CCCP at a final concentration of 25 µg/ml was added to each microtiter plate well containing 2-1024 µg/ml of gentamicin.
- **Comment 3:** *Line 18, why you choose gentamycin, many different antimicrobial agent is more important as anti-staph. Infection. You need to show that study deal with staphylococcus strains that cause skin infection*
- **Response:** Thank you for the nice reminder. As you indicated, gentamicin is commonly employed in the control of post-burn infections and skin infections caused by *S. aureus*. The abstract following your suggestion has been revised on page 1/line 16.
- **Comment 4:** *In this section you need to suggest a possible mechanism for inhibition and is this compound inhibit specific type of efflux pump protein or not.*

- **Response:** Thank you for this suggestion. It would have been interesting to explore this aspect further. However, in the case of our study, it appears to be slightly out of scope as the effects of CGA and CL on efflux pump activity were evaluated phenotypically, and the exact mechanism or level of efflux pump gene expression was not assessed. Based on our study findings, we have observed that these two compounds in combination with gentamicin could effectively inhibit efflux pumps, probably through mechanisms such as covalent binding that impairs pump function, but further research is required to fully understand this.

Introduction:

- **Comment 1:** *In this section you need to paragraph described the family of efflux pump system in staphylococcus and pseudomonas and those that efflux gentamycin.*
- **Response:** Thanks for your comment. In *P. aeruginosa*, intrinsic low-level resistance to aminoglycosides is mediated by the expression of the Mex XY-OprM system. However, as previously mentioned, the objective of our study was to investigate the inhibition of efflux pump activity by evaluating and comparing dye accumulation within the cell using flow cytometry. While it is beneficial to address the issue of which efflux pumps may be involved in gentamicin resistance, incorporating this topic in the introduction may set up expectations for it to be specifically addressed in the results and discussion section, which seems slightly out of the scope of our study.
- **Comment 2:** *Line 57, if you use expression profile (with qPCR or NGS) for selected efflux protein it will good proof for overexpression on down expression of the efflux pump that results from the tested efflux inhibitor*
- **Response:** Thank you for the suggestion. Line 57 of the manuscript provides readers insight into the background of the subject being studied, by referencing earlier research. While the use of techniques such as NGS or qPCR could have offered interesting insights into our study, it was not feasible for us to explore this aspect, because our aim in this study was to investigate efflux pump inhibition using flow cytometry. We used flow cytometry to measure the intracellular accumulation of a fluorescent dye that is a

substrate for the efflux pump, which involved comparing the percentage of cells located in the second quadrant or the gated cells of the test and control. NGS is not readily accessible to everyone in Iran, and taking advantage of its benefits would come at a high cost. While modern techniques offer ease of operation and increased study accuracy, we are unable to utilize them due to the associated high costs.

- **Comment 3:** *Line 67, need to explain role of efflux pump in virulence and aquarrium sensing in bacteria to show how EPIs are important*
- **Response:** Thank you for the nice reminder. The introduction section has been revised accordingly on page 2/lines 61-62.

Results:

- **Comment 1:** *Line 90, you need to mention details about isolated strain, is it from skin or other part, type of infection, host (human or animal).*
- **Response:** Thank you for pointing this out. As mentioned by the reviewer, the revision was made on page 3/ line 93.
- **Comment 2:** *Line 102, basically all EPIs has antimicrobial activity for example Reserpine which well documented as A potent EPIs. Just double check your statement that chromogenic acid and carnosol have no antibacterial activity.*
- **Response:** Thanks for your comment. As was mentioned in the manuscript, on page 4/lines 104-106: “*The antibacterial potentiating effect of chlorogenic acid and carnosol was evaluated. None of the substances exhibited clinically applicable antibacterial activity, as MIC values were above or equal to 1024 µg/ml.*” The antimicrobial effect of the studied compounds in higher concentrations was not evaluated, because of the presence of possible cytotoxicity.
- **Comment 3:** *Line 112, you need to detect the MIC of CCCP also explain why use this inhibitor with gentamicin.*

- **Response:** Thank you very much for pointing this out. The manuscript was revised accordingly. We have made the following changes:
 - ✓ The determination of the MIC of CCCP has been included in the Materials and Methods section on page 10/lines 322 and 326. The results of this determination have been added to the Results section on page 4/lines 115-117.
 - ✓ An explanation for using CCCP has been added to the Materials and Methods section on page 11/lines 354-355.

- **Comment 4:** *Line 123, need to mention some details about isolated strain (host, type of infection etc.)*
- **Response:** Thank you for your comment. Line 123, entitled “*a) Flow cytometric analysis of efflux pump inhibition on the XDR S. aureus-3*” serves as a subheading for the efflux pump inhibition assay within the results section. It may not be suitable to discuss the issue raised by the respected reviewer within this particular section. We previously discussed these details on page 3/line 93.

- **Comment 5:** *Line 125, you need reference " CGA, and CL that did 124 not impact cell viability "*
- **Response:** Thanks for your comment. In line 125, it was mentioned that “*All experiments conducted in this study utilized concentrations of gentamicin, CGA, and CL that did not impact cell viability*” which further points out that sub-inhibitory concentrations of these compounds have been used to continue the study process, “*... to investigate the efflux pump activity, subinhibitory concentrations (1/8, 1/4 and 1/2 MIC) of gentamicin in combination with sub-MIC concentrations of CGA (16, 32, and 64 µg/ml along with 8, 16, and 32 µg/ml) and CL (4, 8, and 16 µg/ml along with 8, 16, and 32 µg/ml) were evaluated, respectively...*”

- **Comment 6:** *Line 139, P. aeruginosa this isolated associated with human infection or not did host receive antibiotic (please provide information about isolates)*

- **Response:** Thank you. Line 139, entitled “*b)Flow cytometric analysis of efflux pump inhibition on the XDR P. aeruginosa-5*” serves as a subheading for the efflux pump inhibition assay within the results section. Only the name of the isolate was mentioned in this part, and the results related to it were discussed further. It may not be suitable to discuss the issue raised by the respected reviewer within this particular section. We previously discussed these details on page 3/line 93.
- **Comment 7:** *Line 162, at least mention that you use cell to check viability with some details*
- **Response:** Thank you for your comment. Line 162, entitled “*Analysis of Cell Viability by the MTT Assay*” serves as a heading within the results section, with the associated results elaborated on subsequently. The details of the MTT assay, including the cell line that was used (NIH/3T3), have been extensively detailed in the Materials and Methods on page 12/lines 396-420.

Discussion:

In this section you need to:

- **Comment 1:** *Explain why use two assumed EPIs*
- **Response:** Thank you. Regarding the use of CGA and CL, it is important to note that these compounds probably possess potential antimicrobial properties and show promise in inhibiting efflux pump activity. Previous studies have investigated this issue in various bacterial isolates or strains that possess specific efflux pumps. As delineated in the discussion section of our study, specifically on page 6/lines 195-202 and page 6/lines 203-215, the significance of CGA and CL has been elucidated.
- **Comment 2:** *Relationship between biofilm and Efflux pump and how EPIs are important*
- **Response:** Thanks for your comment. The relationship between biofilm and efflux pump have been described previously on page 8/lines 244-249.
- **Comment 3:** *Suggested the source of EPIs (plant source or active chemical agent*

- **Response:** Thank you for your comment. In this study, CGA was commercially prepared as an active chemical agent (Merck, Germany), but carnosol was isolated and identified from *Salvia abrotanoides* by Ghanadian et al. and then used. Previously mentioned in materials and methods, page 9/line 300.
- **Comment 4:** *Suggest possible mechanisms for both RPIs that used*
- **Response:** Thanks for your nice reminder. Based on our study findings, we have observed that these two compounds in combination with gentamicin could effectively inhibit efflux pumps, probably through mechanisms such as covalent binding that impairs pump function, but further research is required to fully understand this.
- **Comment 5:** *Is the tested EPIs have selected action or not?*
- **Response:** Thank you for your question. Further research is needed to provide more insight into this issue. Studies should focus on various bacterial isolates and strains, each demonstrating varying resistance to distinct antimicrobial compounds. Nonetheless, the current study demonstrated that CGA and CL potentially exhibit inhibitory effects on XDR *S. aureus* and XDR *P. aeruginosa* isolates, consequently leading to the suppression of efflux pump activity.
- **Comment 6:** *Explain limitation of the study, didn't do expression profile of efflux pump to prof inhibition action of EPIs, used on type of antibiotic*
- **Response:** Thank you for your suggestion. Our main focus was to understand the overall impact of CGA and CL as potential efflux pump inhibitors on antibiotic efficacy, rather than delving into the specifics of efflux pump expression profiles. As mentioned in the conclusion section on page 13, we stated that “...*further advancement to clinical trials is needed to evaluate the feasibility of utilizing CGA and CL for reversing bacterial XDR efflux and determining their efficacy against biofilms...*”.

Methods:

- **Comment 1:** *Line 305, " from patients with wound infection and burns" need to show how isolated the bacteria*
- **Response:** Thank you for pointing this out. Revised accordingly on page 10/line 307-308.

- **Comment 2:** *Line 308, need to mention stage of bacterial growth that used in sensitivity assay.*
- **Response:** Thank you for your nice reminder. Fresh overnight cultures (18-24 h) were used for antibiotic susceptibility testing. Revised accordingly on page 10/lines 312, 318.

- **Comment 3:** *Line 318, it is perfect to used agar-based method to confirm MIC*
- **Response:** Thank you for your comment. Both agar-based and MIC microdilution methods have their unique strengths and limitations. The MIC microdilution method is widely accepted and standardized, making it a reliable and reproducible technique for determining MIC values. It also allows for high-throughput screening of antimicrobial agents.

Re: Spectrum03934-23R1 (Evaluation of Chlorogenic Acid and Carnosol for Anti-Efflux Pump and Anti-Biofilm Activities against Extensively Drug-Resistant (XDR) Strains of *Staphylococcus aureus* and *Pseudomonas aeruginosa*)

Dear Dr. Hossein fazeli:

Your manuscript has been accepted, and I am forwarding it to the ASM production staff for publication. Your paper will first be checked to make sure all elements meet the technical requirements. ASM staff will contact you if anything needs to be revised before copyediting and production can begin. Otherwise, you will be notified when your proofs are ready to be viewed.

Sincerely,
Krisztina Papp-Wallace
Editor
Microbiology Spectrum